# Effect of Ageing in the Mating Behaviour Sequence of *Osmia cornuta* Latr. (Hymenoptera: Megachilidae)

**DOI:** 10.3390/insects14040335

**Published:** 2023-03-29

**Authors:** Antonio Felicioli, Simona Sagona, Francesca Coppola, Chiara Benedetta Boni, Mauro Pinzauti

**Affiliations:** 1Department of Veterinary Sciences, University of Pisa, Viale delle Piagge 2, 56124 Pisa, Italy; 2Interdepartmental Research Centre “Nutraceuticals and Food for Health”, University of Pisa, Via del Borghetto 80, 56124 Pisa, Italy; 3Interdepartmental Centre of Agro-Environmental Research “Enrico Avanzi”, University of Pisa, Via Vecchia di Marina 6, 56122 Pisa, Italy; 4Department of Pharmacy, University of Pisa, Via Bonanno 6, 56126 Pisa, Italy; 5Italian Beekeeping Federation (FAI), Corso Vittorio Emanuele II 101, 00186 Rome, Italy

**Keywords:** mason bee, mating behaviour, antennal courtship, ageing, copula

## Abstract

**Simple Summary:**

The mason bee *Osmia cornuta* is largely used for orchard crop pollination worldwide, both enhancing the maintenance of health ecosystems and providing economic benefits for human society. A deepening of the reproductive biology of *O. cornuta* could refine the management techniques of these pollinators. In the present study, the behavioural sequence of *O. cornuta* mating is described by using Marcov analysis, which revealed the occurrence of repeated, stereotyped, and regular sequences of behavioural units, including two types of copulations, i.e., short and long. The results reported in this paper support the hypothesis that ageing could negatively influence the success of reproduction in mason bees.

**Abstract:**

*Osmia cornuta* Latr. is largely managed worldwide for the pollination of orchard crops, playing a key role in the maintenance of healthy ecosystems and ensuring economic and social benefits for human society. The management techniques of this pollinator include the possibility of delaying emergence from cocoons after diapause, allowing for the pollination of later-blooming fruit crops. In this study, the mating behaviour of bees emerging at the natural time (Right Emergence Insects) and of late-emerged bees (Aged Emergence Insects) was described in order to test if a delay in emergence could affect the mating sequence of *O. cornuta*. Markov analysis of the mating behaviour revealed the occurrence of antenna motion episodes that were repeated in a stereotyped manner at regular intervals during the mating sequence of both Right Emergence Insects and in Aged Emergence Insects. Pouncing, rhythmic and continuous emission of sound, motion of antennae, stretching of the abdomen, short and long copulations, scratching, inactivity, and self-grooming were identified as the stereotyped behavioural units of a behavioural sequence. The occurrence of short copulations, the frequency of which increased with the age of bees, could lead to a failure in the reproduction of the mason bee.

## 1. Introduction

The mason bee *Osmia cornuta* is an obligatory monovoltine solitary bee widespread in Europe, except in the northern countries [1]. This species shows sexual dimorphism [2], and under field conditions, male imagoes emerge one week to two days before females during late winter (February) or early spring (March), depending on the latitude [3,4,5,6,7]. Males patrol the nesting site until mating, which occurs just after the female has emerged and evacuated the meconium [8]. Courtship is performed by the male, and mating usually occurs between a receptive female and only one male [8,9,10]. Nesting activity starts one or two days after female emergence [3,4,5].

*O. cornuta* is a polylectic, gregarious lodger bee that builds allocellous and linear odal nests [11,12]. Embryogenesis, which occurs inside the egg, takes three days, whereupon a first-instar larva hatches. The second-instar larva develops into two transient instars until it reaches the fifth instar larva, which develops firstly in the prepupa and subsequently in the pupa, until the late summer or early autumn, when the mature imago stage develops [13]. The imago enters diapause until spring, when the reproductive cycle can be repeated [12,13,14,15].

Some biological and etho-ecological features of *O. cornuta*, such as the monovoltine life cycle, the gregarious nesting behaviour [16,17], and the polylectic feeding behaviour, have promoted the management of *Osmia* bees as pollinators of orchard crops, both in open fields and in confined environments [18,19]. Moreover, a synchronisation between orchard blooming and the *O. cornuta* flying period occurs [20,21], and this factor, combined with their rapid population growth, contributes to the easy rearing of this pollinator [3,4,5]. In this context, the reproduction of *O. cornuta* is fundamental for a successful output of progeny in rearing [22] and for the maintenance of a healthy ecosystem; furthermore, the efficiency of its pollination services ensures also economic and social benefits for humans [23,24].

Insight into reproduction mechanisms and sexual recognition in *O. cornuta* may support the development of techniques for the rearing and management of solitary bees. Such techniques include the ease of raising *O. cornuta* in artificial nests [3,7,19], the optimisation of the sex ratio in rearing [25,26], the control of diapause duration [27,28], and the possibility of delaying the cocoon emergence after diapause [6,8,29,30,31]. Only female bees perform pollen gathering; the success of courtship and mating behaviour, ensuring diploid egg production, is fundamental for the proper management of *O. cornuta* for pollination purposes [3,4,5,23,24].

In solitary bees, mating systems [32,33,34,35,36,37,38,39,40], the energy cost of reproduction [33,41], fertility conditions [42], territoriality [43], mate guarding [43,44,45], and courtship [46,47,48,49] have been a main focus in evolutive and ecological studies, leading to an increased knowledge of many aspects of the reproductive behaviour of these insects [32,33,34,40,41]. Once a co-specific male and a receptive female come into physical contact, sex recognition, precopulatory courtship, copulation, and postinsemination courtship begin [50]. Antennae, legs, wings, mandibles, and chemical signals are involved singly or in combination [44,47,51,52,53,54]. During male courtship, a rhythmic motion of antennae is very common among Megachilidae [9,36,55], Anthophoridae [56,57], Halictidae [47,53], and Collettidae [56] and can involve both touching and not touching the female antennae. This copulatory behaviour has been well described in studies on *Angochlora pura* [47], *Melissodes tepida* [57], *Lasioglossum zephirum* [47], and *Osmia cornuta* [50].

The mating system of *Osmia* sp. is commonly classified as scramble competition polygyny, and the reproduction success is not altered by female behaviour or choice [37,58] although Conrad et al. [59] found a mating preference expressed by females toward males in *Osmia bicornis.* In the mason bee, the female preference, which also occurs using odour bouquets, appears to be toward males of intermediate size that are capable of producing thoracic vibrations as an honest sign of vigour and health [54,59,60,61].

For an adequate management of *O. cornuta* for both rearing and the pollination of orchard crops, the synchronisation of bee emergence with blossom is fundamental [30]. Synchronisation of bee emergence is usually achieved with the maintenance of cocoons containing diapaused imagoes at 4 °C, until blossoms begin to appear [30,62]. An artificial delay in *Osmia* emergence determines fewer progeny in rearing procedures, and the direct exposure of cocoons to sunlight leads to the immediate death of adults inside cocoons [22,62].

In this work, the hypothesis that delaying the emergence with respect to natural time could affect the mating sequence was tested. To do this, the mating behaviours of naturally emerging bees and of experimentally aged bees were described.

## 2. Materials and Methods

### 2.1. Biological Samples

An experimental population of *O. cornuta* was reared in artificial nests in the Apidology and Proteomics laboratory of the Veterinary Sciences Department of Pisa University. A total of 200 cocoons (n = 100 females, n = 100 males), containing imagoes ready to enter diapause, were collected from artificial nests during autumn and exposed to a temperature of 20 °C, in order to allow all imago bees to reach the diapause stage. After 30 days, cocoons were divided in 2 groups (n = 100 each), and both were placed in climate-controlled chambers at 4 °C and 75% relative humidity. The first group, named Right Emergence Insects (REI), was withdrawn after 77 days, once the potential diapause stage had ended, and 50 males and 50 females were placed in separate Plexiglass cages and exposed to 20 °C until emergence from the cocoons occurs. REI group was composed of 107-day-old mature adult insects (30 + 77 days), which started to emerge from cocoons at the 11th day after exposure at 20 °C and continued to emerge for 13 days, i.e., 11-day interval and 13 days of emergence. The second group, named Aged Emergence Insects (AEI), once a 77-day diapause terminated, was left in climate-controlled chambers (4 °C and 75% relative humidity) for a further 59 days. Cocoons underwent the same protocol as the REI group. AEI group was composed of 166-day-old aged adult insects, which started to emerge one day after exposure at 20 °C and continued to emerge for six days, i.e., one day of interval and six days of emergence. After emergence, bees were fed ad libitum with glucose syrup. Females thereby obtained were collected the day after emergence and placed one at a time in the male cage.

### 2.2. Behavioural Study

In the present study, a behavioural set of time-ordered series of behavioural units in the behavioural sequence of mating in male *O. cornuta* was described in terms of the frequency with which each unit was followed by another, by means of Markov analysis [56].

During the investigation period, 40 pairs of *O. cornuta* for each age group (REI and AEI) performed mating activity. As soon as a male began to climb onto a female, the couple was delicately removed and rapidly inserted into a purpose-built box for video recording. Recordings were made on 120 min VHS Sony (Tokyo, Japan) videocassettes and a Panasonic (Osaka, Japan) F 10 CCD video camera equipped with a Tamron (Saitama, Japan) tele macro 1:3.8/4, 80–210 lens mounted on a tripod and connected to a Panasonic NV-100 video recorder. The box used for recordings was illuminated with photographic-grade quartz lamps (3200 °K). The camera was placed in front of the observation box. Insects were free to move inside the box, and the camera was purposely moved to achieve the best observation angle to record the insertion of genitalia (predominantly 90°, 135°, and 180° on the same plane of the animals).

Recordings were analysed by means of a Panasonic NV-8500 video player equipped with a colour monitor. Videotapes were scanned at different speeds and, when necessary, the observation of single frames was performed. Those videos presenting problems and/or a delay in recording were excluded from mating behaviour analyses. The behaviour in the *O. cornuta* mating sequence was recorded by compiling the continuous-time Markov chain (CTMC), which allows for the calculation of the possibility of the occurrence of a behavioural sequence expressed as a percentage, following [63]. Behavioural categories were chosen in accordance with the principle of parsimony [63] and their intrinsic unitary consistency from the onset to the end.

At the end of each experiment, all tested pairs were removed and released.

## 3. Results

Videotape examination of mating events revealed a passive behaviour in females, while males showed the following action modules, whose sequence was delineated by Markov analysis (Figure 1A–C); the average duration time of each action module is also provided:Pounce (*pounce*): the male flies and rapidly climbs onto the back of the female, encircling the lateroventral portion of her metasoma with both his mesothoracic legs. Duration: from 1 to 2 s.Sound (*sound*): a rhythmic sound is emitted by an oscillation of the anterior legs simultaneously with a slight movement and beating of the wings. Duration: from 30 to 35 s.Antenna motion (*ant*): downward movement of the antennae almost touching the female’s eyes, at first alternating and then simultaneous. Duration: from 30 to 50 s.Buzz (*buzz*): continuous emission of sound obtained by wings vibration. Duration: 2 s.Abdomen (*abdom*): the abdomen is first stretched upward and then toward the ventral part of the female’s abdomen. Duration: from 3 to 28 s.Antennae backward (*back*): the antennae are held forward and with a marked downwards curvature of the distal part of antenna for a duration of 1/24 s, then suddenly return to the starting position. Duration: from 40 s to 1 min.Antenna rotation (*rot*): rapid rotating movement of the antennae in front of the eyes of the female. Duration: from 1 to 2 min.Copulations:
Long copulation (*long*): insertion of the male copulatory organ into the female genital aperture. This copula has duration of 1 or 2 min, during which the male lightly beats his abdomen on the female abdomen.Short copula (*short*): insertion of the male copulatory organ into the female genital aperture, for a duration of only few seconds.
End of copulation (*end cop*): the male withdraws his aedeagus from the female genital aperture.Scratching (*scratch*): this behavioural unit is expressed by the male only following a long copulation, and it comprises three closely connected actions performed in sequence. Duration: from 2 to 6 min.
Simultaneous downward motion of the antennae, which then becomes an alternating movement in an upward direction or an alternating motion both downward and upward.The insect shows the “ant back” movement concomitantly with the emission of a sharp and brief sound and a series of tapping motions performed by curving his abdomen on the urotergal portion of the female.Simultaneous contraction of the meso- and metathoracic legs of the male on the female’s abdomen.
Inactivity (*inact*): the male remains immobile on the back of the female. Duration: from 10 to 20 min.Self-grooming (*groom*): rubbing of the head, the antennae, and the lower part of the thorax using the anterior legs and rubbing of the abdomen by means of the posterior legs, which, in turn, clean each other. Duration: from 2 to 3 min.End of mating (*bye*-*bye*): the male moves away from the female and this action signals the end of mating.

The mating sequence always began with the male pouncing on the female and encircling the lateroventral portion of her metasoma with the mesothoracic legs (*pounce*). *Pounce* was immediately followed by *sound* (100% of observations) and then by a first episode of antenna motion (*ant*, 96%) or *buzz* (4%). *Ant* preceded *buzz* in 98% of cases and *sound* in 2%. In this case, the end of *Ant* partially overlapped with the start of *sound*. *Buzz* preceded *sound*, the first (*ant*) and a second (*ant*1) episode of antenna motion in 11%, 4%, and 85% of cases, respectively. *Ant* occurred without any contact with the female. The stereotyped *ant* action module was repeated four times throughout mating if a long copulation occurred (*ant*1, *ant*2, *ant*3, and *ant*4 in Figure 1A). After an *ant1* episode, the sequence “*abdom*, *back*, and *rot”* followed in the large majority of cases, although in a few cases, the *abdom* module was bypassed to *back* or directly to *rot*. Copulation (*long*) was preceded by *rot* in 100% of cases. During copulation, a new *ant* episode occurred. Between *scratch* and *inact*, an *ant* action module (*ant*3) was very frequently performed. When a further *ant* (*ant*4) followed *inact*, it was very frequently repeated before *groom* and *bye-bye*.

When males performed a short copulation (*short*), they entered a “recovery loop” that was repeated until the occurrence of a long copulation (Figure 1B) or, alternatively, until a last short copulation (Figure 1C). As in long copulation, *ant* (*ant*2 (5)) could precede the end of copulation (*end cop*). *End cop* was followed by *abdom*, instead of *ant*. In a few cases, *abdom* directly led to long copulation and was most frequently followed by *inact* and *ant* (*ant*3 (6)) before the entire sequence began again with *sound*.

In *O. cornuta*, two types of copulatory act, defined as long copulation and short copulation, were observed. In the REI group, a total of 34 mating events occurred, of which 27 consisted of only one long copulation and 7 events consisted of 2 copulations, firstly the short and then the long copulation. The ratio of long-/short-copulation was 4.86, and the long and the short copulation frequencies among all type of copulations was 82% and 18%, respectively. In the REI group, 100% of the mating events ended with a long copulation (Figure 2). In the AEI group, a total of 32 mating events were recorded, of which 5 consisted of only one long copulation; 7 were composed of 2 copulations, firstly the short and then the long copulation; 10 consisted of 3 copulations, firstly two short copulations and then the long one; 5 comprised only 3 short copulations; and 5 consisted of 4 short copulation types (Figure 2). The long and the short copulation frequencies among all types of copulation in the AEI group were 26.19% and 73.8%, respectively. The ratio of long/short copulation was 0.35, and 68.7% of the mating events ended with a long copulation.

Mating consisting of one single long copulation lasted on average 31 min and 30 s, mating in which one short and one long copulation occurred had an average duration of 46 min and 30 s, mating composed of three short copulations lasted on average 68 min and 30 s, and mating consisting of four short copulations had an average duration of 93 min.

## 4. Discussion

A Markov analysis of the mating behaviours in *O. cornuta* revealed behavioural units that were mainly stereotyped, as delineated in the flow ethograms in Figure 1. The behavioural sequence can be divided into four main stages, occurring between the initial attachment and the final detachment. The first stage can be defined as precopulation courtship (from *pounce* to *rot*) and is followed by copulation (*long* or *short*), postcopulation courtship (from *end* to *bye-bye*), and by a potential fourth stage of intercopula looping (ICL), because it occurs as a repetition of the entire sequence and is not followed by *scratch* (stroked *sensu*, [48,59,64]). The first stage is characterised by a slight beating of the wings, giving rise to an intermittent sound [50], concurrent with the rhythmic initially alternating and subsequently simultaneous downward movement of the antennae but without any contact with the female. During this first stage, an oscillating movement of the front legs is also observed. This set of actions may be repeated several times before culminating, with a *crescendo* of excitation, in a continuous emission of sound [50] obtained by the intense wing vibration that is always the prelude to copulation. Male vibration in *O. bicornis*, a species taxonomically close to *O. cornuta*, has been previously well studied, and females were reported to select males that perform longer vibrations, which demonstrates an honest signal of health and vigour [59]. The stereotyped and repeated rhythmic antenna motion in this stage presents extensive similarity with precopulation courtship described for other insects [47,50,53,65,66,67]. This behaviour appears to be linked to intraspecific sex recognition, and induces the female to copulation [47,50]. Behavioural data strongly suggest that male antennas are involved in courtship behaviour. Yin et al. [68] reported the presence of 2-hexyl-1,3-dioxolane in male antennae as well as (*E*)-geranyl acetone in females of *O. cornuta*. Investigations into the role of ageing on the secretion of these two substances during the reproductive behaviour are desirable.

The copula itself may be of the long or short form, as already observed in other studies [9,50]. If long, it has a duration varying from 40 s to 2 min [50]. Because it is always followed by the postcopulatory stage, it can be stated that it determines the end of the mating sequence. Alternatively, when the copulation is short, its duration does not exceed 2 s, and it is followed by a repetition of the entire precopulatory and copulatory sequence (inter-copulation looping (ICL)), which could lead to other short copulations and/or to one last long copulation.

The postcopulatory stage begins at the end of a long copula and is composed of a series of closely linked sequential movements, such as the simultaneous downward movement of the antennae, concomitant contraction of the meso- and metathoracic legs on the abdomen of the female, and downward curving of the abdomen, accompanied by repeated rubbing of the urotergal portion of the female and associated with the emission of a sharp and brief repeated sound. This behaviour is very similar to that described for *O. bicornis* by Raw [36]. It may be a postinsemination association of type c, according to Alcock [44], who reported it only for Coleoptera, Heteroptera, Diptera, and Odonata and not for Hymenoptera.

The stages of precopula, copula, and intercopula looping were also reported by Barrows [47] and Wcislo et al. [53] for Halictidae bees and by Cowan [66] for some eumenid wasps. These authors stated that the precopula stage has a courtship function.

The possibility of repeated events of copulations within a single mating episode, as observed in *O. cornuta* in this study, has also been reported for *O. bicornis* by Seidelmann [37], for *Osmia cornifrons* by Lee et al. [47], and for other hymenopterans [40,49,66]. In *O. bicornis*, repeated events of copulations were reported with a very low frequency [37], without distinction between long and short copula, and using the number of copulations to measure the success of mating [37]. In the present study, the ICL only occurred following a short copulation, i.e., short copulation seems to be the key factor for ICL occurrence. Further information on the duration of copulation in *O. bicornis* could allow for an attempt to rationalise the repeated copulation pattern also present in this species. Conversely, in other Megachilidae, such as *Anthidium* sp., the repeated copulation pattern appears to be the rule, as reported by Alcock et al. [40] and Villalobos and Schelly [49].

In the present experiment, the ICL pattern was observed only in the AEI group (Figure 1B,C), i.e., in bees that were subjected to ageing. In the REI group, consisting of regularly emerged bees, a high frequency of a single long copulations was recorded. Moreover, in the REI group, 100% of mating events ended with a long copulation, and short copulations occurred with a very low frequency, while in the AEI group, only 68.7% of mating events ended with a long copulation. These data suggest that short copulation events became detectable with *O. cornuta* ageing, and that an additional 59 days of exposure at 4 °C determined an increase in reproductive failure of 31.3%. Alternatively, because the time of emergence in *Osmia* decreases with the season and aged males emerged in fewer days [6], the performance of AEI males may have been altered by the reduced available time to establish an adequate mate search behaviour, resulting in a high frequency of short copulations. The association of the ICL pattern and the short copulations occurring only in aged bees and the long copulation always determining the postcopulatory courtship may indicate that long copulation could be the way in which sperm transfer is achieved. Following this hypothesis, the data obtained in this study suggest that aged bees perform less successful mating behaviours than standard-aged bees. Furthermore, an alteration in the duration of diapause can potentially influence the mating success in *Osmia* sp. [14]. Experiments performed by Fliszkiewicz et al. [14] on the filling of the spermatheca suggested that bees subjected to shorter diapause are not in the condition to reproduce, as they inseminated few females. In this context, further investigations on the spermatheca content of aged bees are desirable.

The observation of the mating sequence in this study revealed motionless abdomens of females. However, the lack of movements in females of both the REI and AEI groups during mating behaviour did not necessarily indicate a lack of a role for female choice or biochemical reactions or for reaction to male vibration and size. The key role of pheromones in bee mating behaviours has been the focus of many investigations in recent years [61,68,69]. In this study, olfactory communication was not investigated. For this reason, olfactory differences between REI and AEI females affecting their reproduction in terms of frequencies of long or short copulations could be plausible. Moreover, further investigation into the role of the female in accepting and/or identifying the quality of aged male sperm that could influence the mating sequence is needed.

The results reported in this paper support the hypothesis that ageing in *O. cornuta* causes an increase in short copulation frequency that could lead to a failure of their reproduction. The mason bee is managed worldwide for the pollination of orchard crops; because an artificial shift in its flight period could represent a commercial strategy for the pollination of later blooming fruit crops, further investigations on the success of the two different types of copulation and on aged female fertility conditions that could alter the reproductive success are desirable.

## Figures and Tables

**Figure 1 insects-14-00335-f001:**
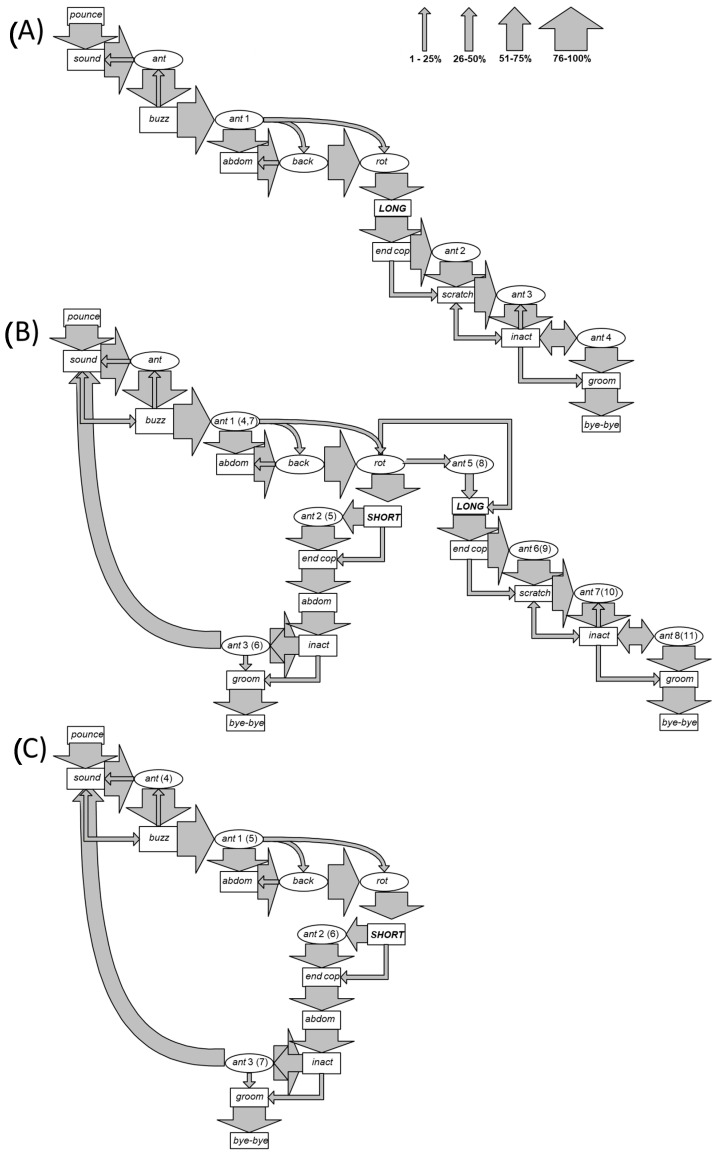
Schematic flow ethograms resulting from the Markov analysis, representing the male mating behaviour of *O. cornuta* Latr. (**A**) Mating sequence including only the long copulation type; (**B**) mating sequence including one or more short copulations (i.e., inter-copula looping: ICL) and a last long copulation; (**C**) mating sequence including only one or more short copulations, i.e., ICL. The thickness of the arrows indicates the probability (expressed in class of percentage) that one behavioural action module precedes the next one.

**Figure 2 insects-14-00335-f002:**
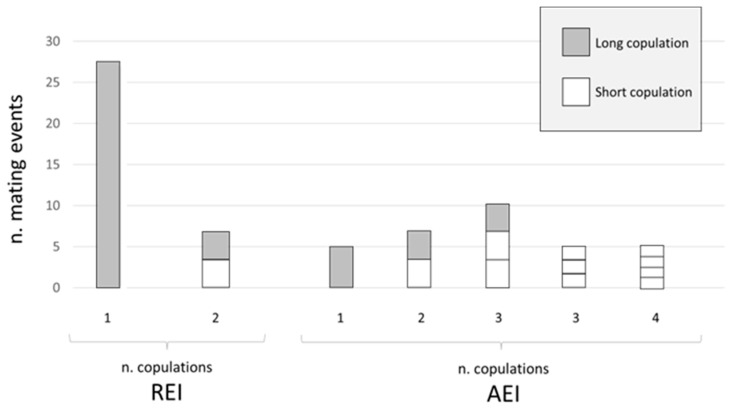
Number of copulations (long in grey; short in white) per number of analysed mating events for *O. cornuta* Right Emerge Insects (REI, on the left) and Aged Emerge Insects groups (AEI, on the right). For mating events composed of more than one copulation, sections in histograms indicate the number and type of copulations, from the first (bottom) to the last (top) occurred, i.e., upper section represents the last copulation.

## Data Availability

Data presented in this study are available on request from the corresponding Author.

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
