# Peer review of "Effect of Ageing in the Mating Behaviour Sequence of Osmia cornuta Latr. (Hymenoptera: Megachilidae)"

_insects, 2023, doi:10.3390/insects14040335_

Round 1

Reviewer 1 Report

The manuscript is very well conceived from a scientific point of view and also gives certain indications for the application of the results obtained for practical purposes. To confirm the assumptions from the discussion and conclusion, it is necessary to continue the research in the direction of practical verification of reproductive success in both groups of bees (REI and AEI). The question arises whether both groups of females are equally successful in leaving offspring (number, sex ratio...)?

Based on the above, I suggest the authors to address this suggestion ( in one sentence) in the last part of the "Discussion" and open the thought for further research in the mentioned direction.

According to the rules, it is sufficient to give the full name of the species (Osmia cornuta) at the beginning of the text and always write it abbreviated (like O. cornuta) in the rest of the text. This should be corrected throughout the text of the manuscript.

The literature has been carefully selected, but I would still suggest adding the following two papers in the "Introduction" section (lines 67-68, where diapause control is written about), keeping in mind that diapause time is essential:

https://doi.org/10.1079/BER2006423 

https://doi.org/10.3390/insects13030235

Reviewer 2 Report

  • There are numerous grammatical errors that need to be fixed. Here are a few examples:

Line 22: … pollination of orchard crops, … OR … pollination in orchards … 

line 23: … human society.

line 25: “later orchards” is an awkward phrasing, I recommend e.g. later blooms or later blooming fruit crops.

line 34: unsuccess is not a noun to my knowledge. Better choice “failure”,e.g.

line 206: …. 18%, respectively.

line 299: … of the two different ….

  • please do not use dashes “-“ (e.g. a week- ten days, line 50) instead of “to” (a week to ten days) throughout the text.
  • line 45: Nesting activity starts one or two days after female emergence.
  • line 66: what do mean by “man-induced”? Please explain in this context.
  • line 70: egg production
  • line 76: … female come into physical contact, … 
  • Line 125-127: Can you provide more information on the underlying concept of the CTMC you compiled as well as the statistical analysis follwing it, as you alluded to in this section? As for the analysis, was a software package involved and if so which one? Please provide more details. 
  • please stick with the code you provided in line 144
  • Line 179-182: this is not traceable from fig. 1A. Please expand or modify the figure in such a way that all 6 ant modules you are referring to are traceable. Is “back” in the text and “ant back” in fig.1A the same? Is “ant1” the same as the “second ant episode”? Do “ant rot” and “rot” refer the same module?
  • Line 183: instead of the word among, between would be a more appropriate word choice here
  • Line 186: “end” or “end cop”. Please stick to your designated codes as given in lines 134 through 172. 
  • Fig. 1: here you consistently write “abdome” which I assume is the same as the module “abdom”. Please be consistent in the useage of the module codes. 
  • What are the percentages based upon (i.e. Probability based on all replicates or probability of occurrence based on the length of a whole sequence)?
  • line 207: no “the” before 100% 
  • line 207: 100% of the mating events in the REI group would be 34 according to line 203. This is not what is shown in fig. 2. Please explain!
  • fig2: why is there a white second bar behind the grey long copulation bar on the left side?
  • lines 207 through 211: numbers 1 through 10 are always written out in text form.
  • figure 1 not 2!
  • this argument only applies to aged male bees, though, correct? As you stated earlier, female bees (aged and regularly emerged ones) remained passive throughout the mating process. Also, do you think it possible that the female influenced the mating sequence through some process unnoticed by you, i.e. it may have found the quality of the sperm of aged male lacking. 
  • Did you test cross-combinations of REI-males+AEI-females and vice versa as well? Would you expect the outcome to be different than the ones you observed? Would you expect such a cross-mating to occur under natural conditions between e.g. aged males and REI females? If not, please outline your reasoning.
  • line 298: When you write, a shift in its flight period, do you mean an artificially (man-made) delay in the emergence of Osmia and subsequent pollination effort? 
  • It is not only the success of the copulation but also (maybe even more so) the fitness of the resulting offspring aside from the offspring sex ratio that ultimately benefits the pollination success in commercial orchards. Do you expect a difference in fitness between REI and AEI offspring? And do you think this would play a role in this context at all? 
  • Overall, the discussion section typically starts by summarizing if your findings support your original hypothesis. This is not the case here as you start out with an observation that is common to both REI and AEI bees, i.e. the passive behavior of the female bees. To my understanding, your main finding is the difference in copulation durations in REI and AEI bees. Please revise this section accordingly. 

Reviewer 3 Report

You address the impact of a prolonged wintering period (not aging) on the mating behavior of mason bee males. That is indeed an important factor for the management of bees. Also the identification of indicators in the behavior of males or females that reliable forecast the result of a mounting would be highly welcome to understand the mate choice in mason bees.

The mating behavior of mason bees has described several times so far for different species. However, mating is very similar in mason bees so your detailed description does not really adds new information to the topic. Moreover, there are several severe concerns that have to list:

1.       “Copulation” is the transmission of sperm from a male to a female. The “short copulations” are simply copulation attempts where the male tries to insert his genitalia. Each mating contains just one copulation, followed by the post-copulatory display.

2.       Females are not passive. The female has to open her sting chamber to allow the male to copulate. Thus, the female decides whether to mate with a given male or not.

3.       There are many factors that could have an impact on (female) mate choice and willingness to accept the courting male. You have completely ignored this fact and biased your study and the interpretation of the results singly to the male behavior.

4.       A Markov analysis is a powerful tool to identify indicators that reliably forecast the outcome of a process. However, you have to analyze the behavior in regard of a successful (in your words “long”) copulations vs. non successful mountings (no post-copulatory display). If you were able to identify such an indicator in the stereotyped mating behavior, than you could analyze in a second step whether and how a prolonged thermal quiescence (the diapause is always the same!) has an impact on that typical behavior.

5.       As the mating behavior is stereotypic sequence of “action modules”, differences are within the modules, that is how intense or long males perform or in the timespan between the modules.

6.       Taina Conrad has nicely demonstrated that vibrations produced during the courtship are essential for the mating success. You have ignored this critical point.

7.       In regard to vibrations, both body size and condition of the male could be important. You have to measure the body size and actual age (days after emergence) for each male and probably also the size relation of male and female.

8.       You describe the mounting and first copulation attempt as if you have videotaped this behavior. However, in the material section you state that couples have been transferred to the observation chamber. As males start a first mating attempt usually immediately after mounting, you could not have analyzed this behavior.

9.       There were 50 females in both groups. In the results you present data of 34 females in REI and only 32 females in AEI. What about the rest? On third of all females is missing!

10.   The hibernation period has a clear impact on the emergence of the bees. The longer the period, the earlier the bees emerge. AND the lag time between males and females becomes shorter. That has several implications for your experiment:

a.       More individuals of the same sex emerge at the same day => You have to handle more females at the same day but you can observe only one couple at a time. Thus, you have more females in the “pipeline” waiting for the experiment in AEI and you measure more females with a delay of two and possibly more days in AEI!

b.      The mean difference between male and female emergence declines with the season => REI males have more time to establish normal mate search behavior in the flight cage. So you perform the experiments with males of different “recovery” phase. That has probably an impact on male performance.

11.   The receptivity of females declines rapidly after emergence. This was demonstrated in one paper you have cited. Thus it makes a great difference whether you tested females at the day of emergence, one day later, or two days later. You have to correct for this bias! Please indicate the age after emergence of the bees for each couple and test for a significant impact.

Some critical points in the experimental setup:

1.       Please indicate the duration between start of incubation and emergence of the bees. This is an indicator of hibernation conditions.

2.       Please indicate the identity and fate of the individual bees. Inseminated females have removed from the experiment. But what about the males? Have there been differences in the mating activity of males? Have some of the males achieved several copulations? Where other males always rejected? Have you tested females not willing to copulate with a given male a second or third time?

3.       You have analyzed the sound production by movements of the legs (!) and wings. However, faint sound can be produced by the males without leg movements or visible vibrations of the wings.

4.       Describe the observation angle of the CCD camera. Have you really been able to observe the insertion of the male genitalia?

5.       The descriptions of the different movements of the antenna are not unambiguous.

6.       The behavior “scratching” is identical to the “post-copulatory display” described in several studies on mason bees.

7.       The durations of the different “action modules” are important as well as the total time from the start of mounting.

Style and preparation of the MS

1.       What is the theoretical framework for the “hypothesis that a delaying of the emergence with respect to natural time could affect the mating sequence”? Are there any implications in the field of behavioral ecology or physiology?

2.       What is the larval development section for? It contributes nothing to the topic.

3.       There are detailed descriptions of the mating behavior of mason bees available in the literature. You do not refer to this body of knowledge in the introduction to deduce your study or to highlight the gap you are going to fill. You should also not ignore the literature body on mate choice (e.g. the work of Taina Conrad) in Osmia bees.

4.       Many of the references (I have checked only a few!) are misquoted! E.g.

a.       [7] deals with parasites and enemies. Patrolling of males and time of receptivity of females are not even mentioned!

b.      [35] describes the mating system of O. bicornis, not the mating behavior.

c.       [57] contains only the suggestion that volatiles might be involved in the mating behavior and cannot be used to prove your statement that this is a fact.

d.      [40] mate guarding has nothing to do with territoriality!

e.      At least one paper dealing with the mating behavior in Osmia is missing.

5.       The references are inflated by many citations that do not really contribute to the topic.

6.       The Markov analysis is simply used to describe the sequence of the “action modules”. That is without a heuristic value as long as you are not able to identify indicators that reliable predict a copulation (sperm transfer).

7.       If you compere REI and AEI mating sequences, where are the differences established by the Markov analysis?

After all, the simple result of the study is presented in Figure 2. In the REI group you observed 34 (long) copulations (100% success) compared to 22 copulations (69% success) in AEI. The reasons for this difference remain unclear as there may be not only differences in male performance but also a reduced receptivity in females. There was not contribution of the Markov analysis to answer this question. And finally, please answer the question raised in the title and indicate the effect of prolonged hibernation on the BEHAVIOR.

Round 2

Reviewer 2 Report

The revisions have greatly improved the manuscript. 

line 45: ... starts one or two days after ... 

line 50: ... into a prepupa and ... into a pupa ...

line 52: ... enters diapause ...

line 61: remove "for human"

line 65: "rearing" instead of "raising"

line 120: ... collected after the day of emergence...

line 133: The camera .... 

line 139: What kind of problems are you referring to here? Could the deliberate exclusion of those (or some of those) videos have biased your analyses and conclusions? 

line 311: ... frequency of single long ....

line 317: ... emerged within a few days ... 

line 338: ... female to either accept .... 

Reviewer 3 Report

You present only minor revisions that do not overcome the methodological problems of the study. And you did not even revise the terminology used (e.g. “short copulations”, “scratching” instead of post-copulatory display). Data on the “history” of your test animals are still missing (time after emergence of females and males). You have added the average duration time of the different action modules but do present an analysis, whether and how the actual duration of a module has an impact on the result of the following mating attempt. There is still no indication from the Markov chain analysis, what module is important for the success of the copulation attempt and how to forecast the result of a courtship. What is the difference in the behavior of successful and unsuccessful males?

The reference list is still inflated by a lot of citations that do not contribute to the theme. You have added some missing references that address the importance of vibrations in Osmia mating, but you do not draw the proper consequences for your study. One paper describing the mating behavior of Osmia bicornis (that is very similar not to say identical to O. cormuta) is still missing. Instead, you cite a Phd-thesis that is not available for readers.

As you have provided only some minor revisions, I could repeat most of the points I suggested in the review of the first version. And last but not least: please answer the question raised in the title and indicate the effect of prolonged hibernation on the BEHAVIOR SEQUENCE.